# Osteophyte Cartilage as a Potential Source for Minced Cartilage Implantation: A Novel Approach for Articular Cartilage Repair in Osteoarthritis

**DOI:** 10.3390/ijms25105563

**Published:** 2024-05-20

**Authors:** Shingo Kawabata, Tomoyuki Nakasa, Akinori Nekomoto, Dilimulati Yimiti, Shigeru Miyaki, Nobuo Adachi

**Affiliations:** 1Department of Orthopaedic Surgery, Graduate School of Biomedical and Health Sciences, Hiroshima University, 1-2-3 Kasumi, Minami-ku, Hiroshima City 734-8551, Japan; shingo7383@gmail.com (S.K.); luckycatleft@gmail.com (A.N.); dilimurat1205@gmail.com (D.Y.); nadachi@hiroshima-u.ac.jp (N.A.); 2Department of Artificial Joints and Biomaterials, Graduate School of Biomedical and Health Sciences, Hiroshima University, 1-2-3 Kasumi, Minami-ku, Hiroshima City 734-8551, Japan; 3Medical Center for Translational and Clinical Research, Hiroshima University Hospital, Hiroshima City 734-8551, Japan; s.miyaki@hiroshima-u.ac.jp

**Keywords:** osteoarthritis, cartilage, osteophyte, chondrocyte, minced cartilage implantation, ACI

## Abstract

Osteoarthritis (OA) is a common joint disorder characterized by cartilage degeneration, often leading to pain and functional impairment. Minced cartilage implantation (MCI) has emerged as a promising one-step alternative for large cartilage defects. However, the source of chondrocytes for MCI remains a challenge, particularly in advanced OA, as normal cartilage is scarce. We performed in vitro studies to evaluate the feasibility of MCI using osteophyte cartilage, which is present in patients with advanced OA. Osteophyte and articular cartilage samples were obtained from 22 patients who underwent total knee arthroplasty. Chondrocyte migration and proliferation were assessed using cartilage fragment/atelocollagen composites to compare the characteristics and regenerative potential of osteophytes and articular cartilage. Histological analysis revealed differences in cartilage composition between osteophytes and articular cartilage, with higher expression of type X collagen and increased chondrocyte proliferation in the osteophyte cartilage. Gene expression analysis identified distinct gene expression profiles between osteophytes and articular cartilage; the expression levels of COL2A1, ACAN, and SOX9 were not significantly different. Chondrocytes derived from osteophyte cartilage exhibit enhanced proliferation, and glycosaminoglycan production is increased in both osteophytes and articular cartilage. Osteophyte cartilage may serve as a viable alternative source of MCI for treating large cartilage defects in OA.

## 1. Introduction

Articular cartilage defects occur in association with joint trauma, including sprains, dislocations, and fractures, and can eventually lead to osteoarthritis (OA), along with other factors such as work, sports participation, musculoskeletal injuries, obesity, and gender [1]. Autologous chondrocyte implantation (ACI) is a surgical treatment for focal cartilage defects with satisfactory outcomes [2]. Although ACI is a well-established procedure for focal cartilage defects, it remains challenging for cartilage defects in OA, particularly larger defects [3]. In the ACI procedure, harvesting normal articular cartilage is required. However, there is a lack of normal cartilage that serves as a source of chondrocytes in patients with advanced OA. Moreover, in relatively young patients with traumatic OA, the remaining normal cartilage should be preserved as much as possible. Therefore, a new method of cartilage regeneration that does not require harvesting normal cartilage is necessary. Additional disadvantages of ACI include two-step surgeries to harvest the cartilage and implant the cultured chondrocytes, and time and cost consumption of chondrocyte culture, although ACI can repair cartilage defects with hyaline cartilage.

Minced cartilage implantation (MCI) has been developed as a one-step surgery to overcome the disadvantage of ACI [4]. In this approach, the harvested cartilage is minced into 1 to 2 mm pieces and dispersed onto a biodegradable scaffold. The composite is then implanted into the cartilage defect. Several animal studies, including histological findings, have demonstrated comparative outcomes between MCI and ACI [5,6]. This method also resulted in satisfactory outcomes clinically [7,8,9]. Therefore, MCI is a promising cartilage repair method as a one-step procedure for large cartilage defects. However, similar to ACI, MCI is limited by the amount of normal cartilage in non-weight-bearing areas that serve as a source of chondrocytes in advanced OA. Therefore, other cartilage tissues must be harvested as alternatives to the normal articular cartilage to perform MCI for large cartilage defects.

OA has various features including osteophyte formation, osteosclerosis of the subchondral bone, and cartilage degeneration [10]. Osteophytes typically form at joint margins as osteocartilaginous outgrowths and can cause pain and functional impairment. Osteophytes are established through the growth of an initial cartilage template that is partially replaced by bone marrow cavities, and the bone is typically covered by a cartilage cap that can merge with articular cartilage at later stages [11]. Osteophytes should be removed during OA treatment, and the cartilage within these osteophytes can potentially be a source of chondrocytes for MCI. Previously, the differences in gene expression between articular and osteophyte cartilages have been studied to examine the feasibility of osteophyte cartilage as a source of chondrocytes [12,13]. However, to the best of our knowledge, no reports have evaluated the cell migration, proliferation, and matrix production from cartilage fragments in scaffolds for MCI application. We hypothesized that osteophyte chondrocytes have the capacity for cell migration, proliferation, and matrix production, which are important for cartilage repair, and are comparable to articular cartilage, although the properties of osteophyte cartilage differ from those of articular cartilage. The present study aimed to evaluate the differences between osteophytes and articular cartilage, especially the migration, proliferation, and matrix production capacities for the feasibility of MCI using osteophyte cartilage.

## 2. Results

### 2.1. Osteophyte Cartilage Has Similar Properties to Articular Cartilage

In osteophytes, a cartilage layer with strong safranin -O staining covered by a fibrocartilaginous layer with less intense safranin -O staining and type I collagen positivity was observed. Beneath the cartilage layer, bone structures, such as the subchondral bone plate and underlying bone marrow cavity, were observed; however, no tidemark was visible. In articular cartilage, the superficial layer was less stained with safranin -O, while the middle and deep layers showed strong safranin -O staining. Type II collagen was strongly stained in the cartilage layer of osteophytes and articular cartilage, indicating high proteoglycan contents. Chondrocytes expressing type X collagen were mainly distributed in the deep layers of the osteophytes and articular cartilage; however, these cells were more widely distributed in the osteophyte cartilage (Figure 1A). Semi-quantitative analysis using IOD/area showed that type I and II collagen did not differ significantly between osteophytes and articular cartilage, and the integrated optical density (IOD)/area of type X collagen in osteophyte cartilage was significantly higher than that in articular cartilage (*p* < 0.01) (Figure 1B). Ki67-positive cells in the osteophyte cartilage were distributed throughout the cartilage layer and were abundantly expressed in chondrocytes within clusters. In articular cartilage, Ki67-expressing cells were more abundant in the superficial layers, whereas proteoglycans were decreased (Figure 1C). The Ki67-positive cell ratio in the superficial layer did not differ significantly between the osteophyte and articular cartilages; however, the Ki67-positive cell ratio in the middle and deep layers was significantly higher in the osteophyte cartilage than in the articular cartilage (*p* < 0.01) (Figure 1D).

Bulk RNA sequence analysis revealed that 11,705 genes were up-regulated and 4702 genes were down-regulated in osteophyte cartilage compared with articular cartilage. Among these genes, four of the up-regulated genes (COL1A1, COL4A2, OLFML3, and BASP1) and five of the down-regulated genes (BMP2, S100B, SMOC2, ITM2A, and MGP) had padj < 0.05 (Figure 2). There was no difference in the expression of COL2A1, ACAN, and SOX9 between the osteophytes and articular cartilage (Table 1).

### 2.2. Matrix Staining of Osteophyte and Articular Cartilage Fragments in Atelocollagen Gel Is Maintained after 6 Weeks of Culture

Composites using atelocollagen gels with the minced cartilage or isolated chondrocyte were divided into six groups (Table 2). In the MO1 and MO2 groups, the cartilage fragments in the gel were stained with safranin -O after 3 weeks of culture. However, the intensity of safranin -O decreased after 6 weeks of culture (Figure 3A). In the MC1 and MC2 groups, cartilage fragments were well stained with safranin -O after 3 weeks of culture, and staining reduction after 6 weeks of culture was milder than that in the MO groups (Figure 3B). In the IC and IO groups, dispersed chondrocytes were observed in the atelocollagen gel, and no staining of safranin -O was observed (Figure 3C). At 3 weeks, the Bern scores in the MC and MO groups were significantly higher than those in the IC and IO groups (*p* < 0.01). The Bern scores in MC2 and MO2 were significantly higher than those in MC1, and those in the MC1 and MC2 groups were significantly higher than those in the MO1 group (*p* < 0.01). In addition, the Bern score of the MO2 group was significantly higher than that of the MC2 group (*p* < 0.01). At 6 weeks, the results were the same as those at 3 weeks, except that the difference between MO2 and MC2 was no longer significant. From 3 to 6 weeks of culture, the Bern score showed no significant differences in all groups, except for the IO group, in which the Bern score significantly increased (*p* < 0.05) (Figure 3D). The scores of matrix staining of cells in MO1 and MO2 were significantly higher than those in MC1 and MC2, respectively; however, there were no significant differences between MC1 and MO1, and MO1 and MO2 at 6 weeks (Figure 3E). Bone formation was not observed in any of the groups.

### 2.3. Osteophyte Cartilage Has Better Cell Migration and Proliferation Abilities in the Gel than Articular Cartilage

To evaluate the ability of chondrocytes to migrate from the cartilage, the number of cells in each group was calculated in the gel. In the IC and IO groups, the cells accumulated at the edge of the gel and were sparse inside the gel (Figure 4A). However, the number of cells in the IO group was significantly higher than that in the IC group at 3 and 6 weeks (*p* < 0.01) (Figure 4B). All groups of cartilage fragments embedded in the gel showed migrating cells around the cartilage fragments in the gel after 3 weeks of culture. At 6 weeks, the number of cells in the gels increased and even accumulated at the edge of the gel (Figure 4A). Both the MO1 and MO2 groups had significantly more cells in the gel than the MC1 and MC2 groups at 3 and 6 weeks (*p* < 0.01) (Figure 4B).

### 2.4. Migrated and Proliferated Cells in the Gel Are Chondrocytes from Osteophyte Cartilage

To determine whether the cells migrating into the gel were chondrocytes, the expression of LECT1 (chondromodulin), a cartilage-specific protein, was evaluated. LECT1-positive cells were observed in all groups at 3 and 6 weeks (Figure 5). At 3 weeks, the LECT1-positive cell rate was higher for MC2 than for MC1 and for MO2 than for MO1, indicating that more cartilage fragments promote the migration of more chondrocytes in the gels (Figure 5D). At 6 weeks, the number of LECT1-positive cells did not differ significantly among the MC1, MO1, MC1, and MC2 groups (Figure 5D).

### 2.5. Osteophyte Chondrocytes Have GAG Production Ability and Better Proliferation Potential than Articular Chondrocytes

Cell proliferation assays revealed that the cell proliferation of chondrocytes from osteophytes significantly increased compared with that of chondrocytes from articular cartilage at 48 and 72 h (*p* < 0.01 for both) (Figure 6A). At 3 and 6 weeks, all groups had GAG content, and this was significantly higher in the MC2 group than in the MO2 group (*p* < 0.05 and *p* < 0.01, respectively); however, there was no significant difference in the GAG contents between the MC1 and MO1 groups. At 3 and 6 weeks, the GAG contents in the MC1, MO1, MC2, and MO2 groups were significantly higher than those in the IC and IO groups. From 3 to 6 weeks, GAG contents increased in all groups: IC, 1.9 ± 0.7-fold; IO, 2.6 ± 1.7-fold; MC1, 1.4 ± 0.5-fold; MO1, 1.2 ± 0.7-fold; MC2, 1.6 ± 1.3-fold; and MO2, 1.1 ± 0.4-fold (Figure 6B).

## 3. Discussion

This study revealed that osteophyte cartilage has articular cartilage features, such as proteoglycan content, and its chondrocytes have greater cell proliferation ability than articular chondrocytes, although osteophyte cartilage also has more bone elements, such as COLX, compared to articular cartilage. Minced osteophyte cartilage/atelocollagen composites showed chondrocyte migration from the fragments and GAG production in the gel, suggesting that the implantation of this composite into articular cartilage defects could be a novel cartilage regeneration procedure for patients with OA.

The concept of MCI was first reported in 1982 by Albrecht et al. [14], and an animal study by Lu et al. in 2006 demonstrated that cartilage fragments are a useful cell source for cartilage repair [5]. Although the concept of MCI is not new, it has received increasing attention in recent years owing to its single-step procedure, strong biological potential, and high cost-effectiveness [4,15]. An animal study by Matsushita et al. demonstrated that MCI yields well-repaired tissue, comparable to that of ACI [6]. Clinically, it is reported that patients undergoing MCI had satisfactory outcomes [7,8,9]. For cartilage defects associated with OA, ACI has been performed and allowed a delay in arthroplasty [3]. In OA joints, increased inflammation and catabolic processes may increase the risk of treatment failure [16]. Using an in vitro inflammation model, Ossendorff et al. demonstrated that MCI has superior regeneration potential in OA conditions compared to ACI because MCI is less susceptible to inflammatory cytokines with reduced IL-6 release [17]. MCI using osteophyte cartilage has the potential to be a more effective OA treatment option compared to ACI.

Few studies have reported on the molecular characterization of osteophyte cartilage as a potential source for cartilage repair. In a report by Gelse et al., mature osteophytes largely resembled articular hyaline cartilage with a predominance of COL2 and aggrecan [18]. Another report by Gelse et al. revealed molecular differences in chondrocytes between osteophytes and articular cartilage using a microarray [12]. They showed that osteophyte chondrocytes had increased expression of genes related to the endochondral ossification, including BGLAP, BMP8B, COL1A2, SOST, GADD45β, and RUNX2, and genes related to tissue remodeling enzymes, including MMP9, MMP13, and HAS1. In contrast, articular chondrocytes showed increased expression of genes related to antagonists and inhibitors of the BMP- and Wnt-signaling pathways, including GREM1, FRZB, and WISP3, and of genes related to the inhibition of terminal chondrocyte differentiation and endochondral bone formation, including PTHLH, SOX9, STC2, S100A1, and S100B. Our study revealed that osteophyte cartilage had a higher expression of type X collagen than articular cartilage. Previous reports demonstrated that COLX is predominantly located around hypertrophic and clustered chondrocytes [19], and COLX expression is elevated in human OA cartilage as a result of chondrocyte hypertrophy and cartilage calcification [20]. Thus, the osteophyte cartilage has more bony features than the articular cartilage, although the osteophyte cartilage has obvious cartilaginous tissue. In addition, chondrocytes derived from osteophyte cartilage have a greater potential for proliferation than those derived from articular cartilage. This difference was further examined using RNA-sequencing, with four genes (COL1A1, COL4A1, OLFML3, and BASP1) and five genes (BMP2, S100B, SMOC2, ITM2A, and MGP) found to be up- and down-regulated, respectively, in osteophyte cartilage compared to articular cartilage. COL1A1 expression in osteophyte cartilage can reasonably be considered higher than that in articular cartilage because COL1A1 is a marker of bone and cartilage degeneration. COL4A1 is essential for the stability and function of the vascular basement membrane [21]. COL4A1 also forms a heterotrimer with COL4A2, which promotes osteogenic differentiation through negative regulation of the Wnt/ β-catenin pathway [22]. Angiogenesis is an important factor in bone formation, and OLFML3 accelerates neovascularization by promoting endothelial cell proliferation and migration [23]. BASP1 is a negative regulator of RANKL-induced osteoclastogenesis, indicating that BASP1 up-regulation may shift osteogenesis [24]. Among the down-regulated genes, S100B and SMOC2 enhance inflammation in arthritic conditions [25,26], suggesting that MCI using osteophyte cartilage may be more resistant to inflammation. MGP prevents calcification by regulating BMP2, thereby modulating osteoinduction [27]. ITM2A is expressed at the onset of chondrocyte differentiation in growth plates, and inhibits the early stages of chondrogenic differentiation of mesenchymal stem cells [28,29]. BMP2 also has chondrogenic effects [30], and decreased expression of these genes leads to a bone phenotype. However, COL2A1, ACAN, and SOX9 expression did not differ significantly between the osteophytes and articular cartilage. Although osteophyte cartilage contains more bone elements than articular cartilage, it contains important elements, such as cartilage, at the same levels. LECT1-positive cells were observed in the gel and their number increased in the composite prepared using minced osteophyte cartilage. Moreover, GAG was found in the gels, and its production increased at 6 weeks in the composite prepared using minced osteophyte cartilage. Osteophyte cartilage is a useful cell source for articular cartilage repair using a composite with atelocollagen gel for cartilage defects in OA. In addition, subchondral bone condition during cartilage repair in OA must be improved [31]. Cartilage properties in addition to bone elements in the osteophyte cartilage may contribute to osteochondral unit regeneration in OA.

Patients with OA typically have large areas of cartilage defects that require treatment. Compared with the composites of suspended chondrocytes from enzymatic digestion, the composites of minced cartilage from both osteophytes and articular cartilage contained significantly more chondrocytes in the gel, suggesting that enzymatic digestion damaged chondrocytes. Based on our preliminary data, 100 mg of cartilage fragments contained approximately 2 × 10^5^ chondrocytes [32]. In this study, the isolated chondrocyte composite contained cells isolated from 100 mg cartilage fragments in 100 μL gel. The chondrocyte migration and Bern score in the group containing 25 mg of minced cartilage in 100 μL of atelocollagen gel were superior to those in the isolated chondrocyte groups, which means that the amount of cartilage in the minced cartilage procedure is only one-fourth that of isolated chondrocytes. Thus, minced cartilage embedded in atelocollagen gel can potentially cover a cartilage defect 4 times larger than that covered by conventional ACI. In particular, the cell proliferation ability in the osteophyte cartilage was greater than that in the articular cartilage, suggesting that MCI using osteophyte cartilage could cover a larger area than that using articular cartilage. MCI using osteophyte cartilage may be an alternative option for restoring the articular surface in OA.

This study had several limitations. First, the cartilage tissues were harvested from predominantly elderly women patients with OA who underwent total knee arthroplasty (TKA). A nationwide cohort study, Research on Osteoarthritis Against Disability (ROAD), revealed that radiographic OA was present in 47.0% and 70.2% of men and women, respectively [33]. Traumatic OA can occur in young patients who require articular cartilage regeneration to preserve joint function. The structural and mechanical properties, including the molecular aspects of the cartilage, change with aging [34]. The evaluation of osteophytes and articular cartilage in young populations is also required. Second, synovial cell contamination is possible in the osteophyte cartilage groups. However, approximately 90% of the cells in the gel of the osteophyte groups were LECT1-positive, which was comparable to the share in the articular cartilage groups. Even if synovial cells are contaminated, they are favorable for cartilage regeneration due to the high chondrogenic capacity of synovial stem cells [35]. Finally, whether osteophyte cartilage will become bone in the long term is unclear, because our evaluation only lasted for 6 weeks of culture. Because the implanted composite is influenced by the environment of the implanted site, an in vivo study in which a composite with osteophyte cartilage is implanted into the cartilage defect should be conducted and evaluated over a long period. Further studies are required to address these limitations.

## 4. Materials and Methods

### 4.1. Participants

Twenty-two patients, comprising seven men and fifteen women, who underwent TKA, were enrolled in this study. The patients had a mean age of 76.6 ± 6.9 (range, 62–85) years. All patients were diagnosed with primary OA of Kellgren–Lawrence grade 3 or 4 and varus alignment. Patients with secondary OA, systemic joint diseases such as rheumatoid arthritis, or valgus alignment were excluded. During TKA, osteophytes at the edge of the medial epicondyle of the femur, classified as stage 4 as previously described [18], were harvested. After bone resection, the resected osteochondral tissue was obtained from the lateral posterior condyle of the femur. Osteophyte and articular cartilages from 6 specimens of 22 patients, with a mean age of 79.5 ± 4.3 (71–82) years, were histologically evaluated, and another 5 specimens, with a mean age of 68.6 ± 8.4 (62–80) years, were used to analyze gene expression. The remaining specimens from 11 patients with a mean age of 78.5 ± 4.5 (72–85) years were used to prepare the cartilage composite. This study was approved by the local ethics committee of our university, and informed consent was obtained from all participants.

### 4.2. Preparation of Cartilage/Atelocollagen Composite

Cartilage/atelocollagen composites were prepared according to a previous report [31]. Cartilage was removed from the subchondral bone using a scalpel. The cartilage was then washed in 0.9% sodium chloride and minced manually to obtain cartilage fragments of <1 mm^3^. To isolate chondrocytes, cartilage fragments were treated with 0.25% trypsin (Gibco, Carlsbad, CA, USA) in sterile saline for 30 min, followed by 0.25% collagenase type 2 (Gibco, Carlsbad, CA, USA) in Dulbecco’s modified Eagle’s medium (DMEM; Gibco, Carlsbad, CA, USA) supplemented with 10% fetal bovine serum (FBS; Sigma, Taufkirchen, Germany) and antibiotics (penicillin [10,000 units] and streptomycin [10,000 μg/mL] (Nacalai tesque, Kyoto, Japan) for 4 h at 37 °C in a culture tube, according to methods previously described [36]. The chondrocytes were washed 3 times with culture medium and then filtered through a 70 mm sterile nylon mesh (Cell Strainer, BD Biosciences Discovery Labware, Franklin Lakes, NJ, USA). Isolated chondrocytes were used to prepare the composite and for cell proliferation assays. Isolated chondrocytes (2.0 × 10^5^ cells) from articular (isolated chondrocyte from articular cartilage: IC group) and osteophyte cartilages (isolated chondrocyte from osteophyte: IO group) were dispersed and mixed in 100 μL of atelocollagen gel (Koken, Tokyo, Japan). Cartilage fragments from the articular (minced cartilage: MC group) and osteophyte cartilages (minced osteophyte: MO group) were embedded in the atelocollagen gel. These groups were divided into two groups according to the volume of minced cartilage mixed in atelocollagen gel (12.5 mg: MC1 and MO1 groups, 25 mg: MC2 and MO2 groups) (Table 2). All cell and minced cartilage mixtures were placed in 6-well plates and incubated in a mixture of 5% CO2 and 95% air at a temperature of 37 °C for 3 and 6 weeks. The culture medium was changed every 3 days, and L-ascorbic acid (50 μg/mL) was added every 2 days.

### 4.3. Histological Analysis

Samples for histological analysis of articular and osteophyte cartilages were immediately fixed in 4% paraformaldehyde (PFA; Wako Pure Chemical Industries Ltd., Osaka, Japan). The samples were decalcified in EDT-X for three weeks. They were embedded in paraffin and sliced into 5 μm thick sections. The sections were stained with safranin -O Fast Green.

After 3 or 6 weeks of incubation, atelocollagen composites were fixed in 4% PFA, at 4 °C overnight, and subsequently embedded in paraffin. Four-micrometer sections were prepared and stained with safranin -O Fast Green and hematoxylin/eosin. Each sample was evaluated using the Bern score [37]. The Bern score (minimum score, 0; maximum score, 9) is based on 3 items: uniformity and intensity of safranin -O staining, distance between cells/amount of matrix produced, and cell morphologic characteristics. Each item is scored from 0 to 3. To evaluate chondrocyte migration and proliferation in the atelocollagen gel, 6 areas (500 μm × 500 μm grid) were randomly selected in each section and cells were counted under a magnification of ×400. To evaluate cartilage degeneration of the fragment in the gel, subscales of the modified Mankin Score, cells (0–3 points), and matrix staining (0–4 points), were used [38].

### 4.4. Immunohistochemistry

Immunostaining each section was performed using anti-collagen type I antibody (Abcam, Cambridge, UK, #ab138492, 0.67 μg/mL), anti-collagen type II (DSHB, Iowa, IA, USA, #II-II6B3-C, 5 μg/mL), anti-collagen type X (DSHB, Iowa, USA, #X-AC9-S, 66.7 μg/mL), anti-Ki67 antibody (Thermo Fisher Scientific, Waltham, MA, USA, #8D5 MA5-15690, 2.5 μg/mL), and anti-LECT1 antibody (Abcam, Cambridge, UK, #ab714501, 20 μg/mL). Slides of anti-collagen type I, anti-Ki67, and anti-LECT1 antibodies were pretreated with antigen-retrieval reagent (Immunoactive; Matsunami Glass Ind, Ltd., Osaka, Japan) at 60 °C for 16 h, followed by blocking serum for 30 min. Slides of anti-collagen type II, Ⅹ antibody were pretreated with pretreated with Proteinase K, followed by blocking serum for 30 min. The sections were immunostained with anti-collagen type I, II, X, and Ki67 antibodies diluted in Can Get Signal immunostaining solution (TOYOBO, Tokyo, Japan). The sections were visualized using the avidin–biotin system (Vectastatin Elite ABC Mouse IgG kit, Vector Laboratories, Inc., Burlingame, CA, USA) and 3,3′-diaminobenzidine (Peroxidase Substrate Kit, Vector Laboratories, Inc.), according to the manufacturer’s instructions. For semi-quantitative analyses of collagen type I, II, and X expressions, the data were expressed as the average integrated optical density (IOD/area), according to a previous report [39]. Three areas (300 × 300 μm grid) in the cartilage layer of each section were randomly selected and cells were counted under 400× magnification. To quantitatively evaluate the Ki67 expression, three 300 × 300 μm squares were randomly set in the cartilage layer and the total cell number, and the number of Ki67-positive cells, were counted respectively. Then, the percentage of positive cells was calculated. For the LECT1, Alexa Flour 488-conjugated anti-rabbit IgG (Molecular Probes, Invitrogen, Carlsbad, CA, USA, 5 μg/mL) as a secondary antibody was used after immunostaining with the anti-LECT1 antibody. A DAPI (4′6-diamidino-2-phenylindole; Dojindo Laboratories, Kumamoto, Japan) solution was applied for nuclear staining. Three 300 × 300 μm squares were randomly set in the gel and the total cell number and the number of LECT1 positive cells were counted respectively. The percentage of positive cells was calculated.

### 4.5. Cell Proliferation Assay

Cell proliferation assays were performed using Cell Counting Kit-8 (CCK-8; Dojindo Laboratories, Kumamoto, Japan). Chondrocytes isolated from osteophytes and articular cartilage were seeded in 96-well plates at a density of 5 × 10^3^ cells per well and cultured for 72 h in DMEM containing 10% FBS. The substrate (highly water-soluble tetrazolium salt; WST-8) was then added to each well. After one hour of incubation, absorbance was measured at 450 nm using a microplate reader at 24, 48, 72, and 94 h.

### 4.6. Evaluation of GAG Contents

After 3 and 6 weeks of incubation, half of the composites in each group were used for biochemical assays for GAG quantification using the Blyscan Glycosaminoglycan assay kit (Biocolor, Carrickfergus, UK), according to the manufacturer’s protocol.

### 4.7. RNA Sequence

Total RNA was extracted from the minced cartilage of osteophytes and articular cartilage using the Isogen reagent (Nippon Gene, Tokyo, Japan) and an RNA purification kit (Direct-Zol RNA MicroPrep, Zymo Research, Irvine, CA, USA). RNA sequence libraries were prepared using QuantSeq 3′mRNA-Seq library Prep Kit using single-end 75 base read sequencing using an Illumina NextSeq500 sequencer. After checking the quality of the reads, filtering low-quality reads was found to be unnecessary. The reads were mapped to the mouse reference genome (mm10), and the expression of the identified genes was normalized by calculating TPM. Differentially expressed genes were extracted and subsequently imported into gene ontology enrichment analysis using Metascape (https://metascape.org) [40].

### 4.8. Statistical Analysis

Statistical analyses between two groups were performed using the paired *t*-test, and multiple comparisons were performed using the Steel–Dwass test. The data are presented as mean ± standard deviation (mean ± SD). Statistical significance was set at *p* < 0.05. Statistical analysis was performed in GraphPad Prism 9.0 (San Diego, CA, USA).

## 5. Conclusions

This study demonstrates that osteophyte cartilage is a useful chondrocyte source in composites with atelocollagen for minced cartilage implantation. While joint preservation surgery has garnered much attention, MCI using osteophyte cartilage is an alternative option for cartilage regeneration in patients with OA, without sacrificing the remaining normal articular cartilage.

## Figures and Tables

**Figure 1 ijms-25-05563-f001:**
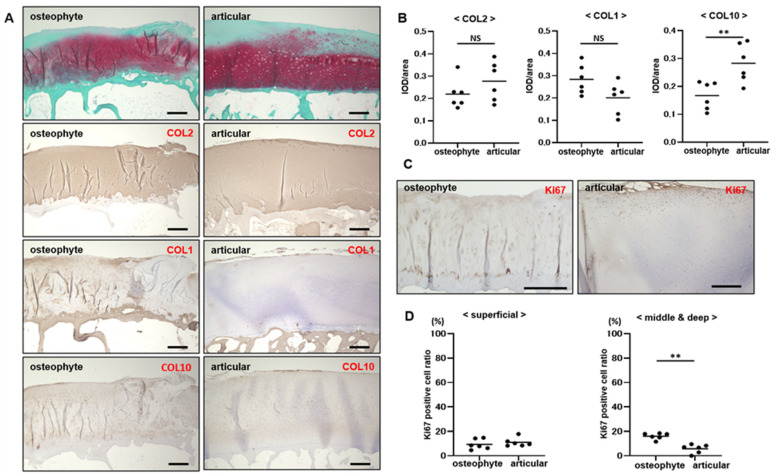
Histological analyses between osteophyte and articular cartilage. (**A**) Safranin -O/Fast green staining and immunohistochemistry of type II, I, and X collagen. (**B**) IOD/area of type II, I, and X collagen in the immunohistochemistry. (**C**) Immunohistochemistry of Ki67. (**D**) Ki67-positive cell ratio. Bar indicates 500 μm. NS—no significant difference. **—*p* < 0.01.

**Figure 2 ijms-25-05563-f002:**
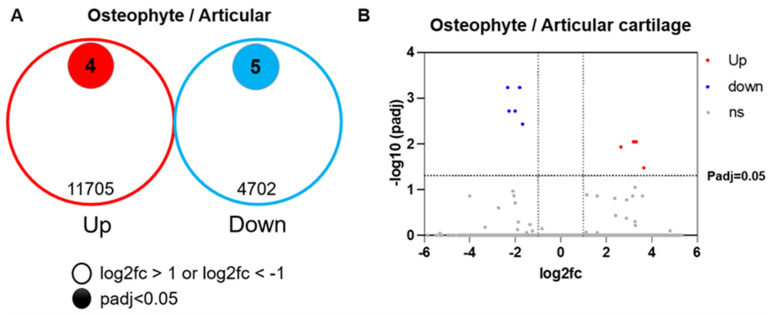
Gene expression profile comparing osteophyte and articular cartilages. (**A**): Up- and down-regulated genes of osteophyte cartilage to articular cartilage. A total of 11,705 genes with log2fc > 1 were up-regulated and 4 genes with padj < 0.05 were identified; 4702 genes with log2fc < −1 were down-regulated and 5 genes with padj < 0.05 were identified. (**B**): Volcano plot of differentially expressed genes with padj < 0.05. ns—no significant difference.

**Figure 3 ijms-25-05563-f003:**
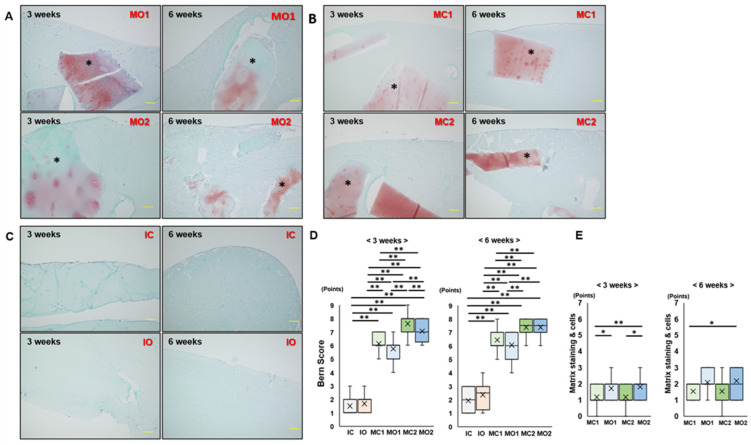
Histological evaluation of atelocollagen composite by safranin -O/Fast green staining. (**A**) MO1 (minced osteophyte cartilage; 12.5 mg) and MO2 (minced osteophyte cartilage; 25 mg) groups. (**B**) MC1 (minced articular cartilage; 12.5 mg) and MC2 (minced articular cartilage; 25 mg) groups. (**C**) IC (isolated articular cartilage chondrocyte) and IO (isolated osteophyte cartilage chondrocyte) groups. *—cartilage fragment. Bar indicates 100 μm. (**D**) Bern score in each group at 3 and 6 weeks. (**E**) Matrix staining and cell scores for evaluation the cartilage fragment at 3 and 6 weeks of culture. *—*p* < 0.05. **—*p* < 0.01. The line of the box—median. ×—mean. N = 11 in each group.

**Figure 4 ijms-25-05563-f004:**
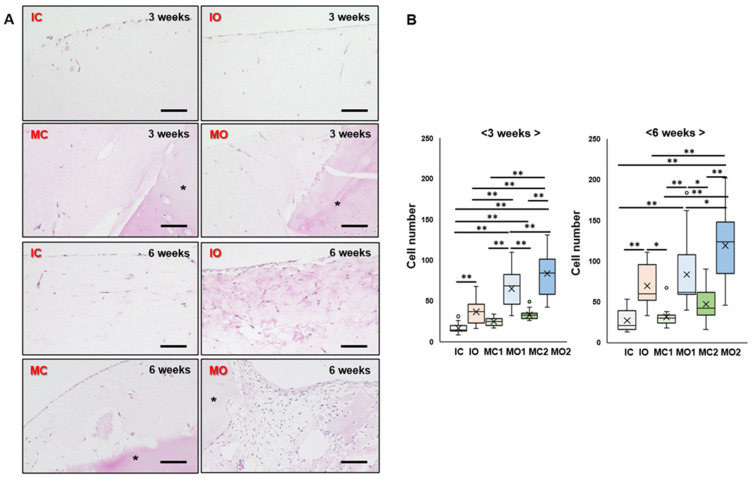
Cell migration in the atelocollagen gel after 3 and 6 weeks of culture. (**A**) Representative images of isolated chondrocytes and cartilage fragments from articular and osteophyte cartilages stained using hematoxylin eosin. *—cartilage fragment. Bar indicates 100 μm. (**B**) Cell number in the gel at 3 and 6 weeks of culture. *—*p* < 0.05. **—*p* < 0.01. The line of the box—median, ×—mean, small circle—outlier. N = 11 in each group.

**Figure 5 ijms-25-05563-f005:**
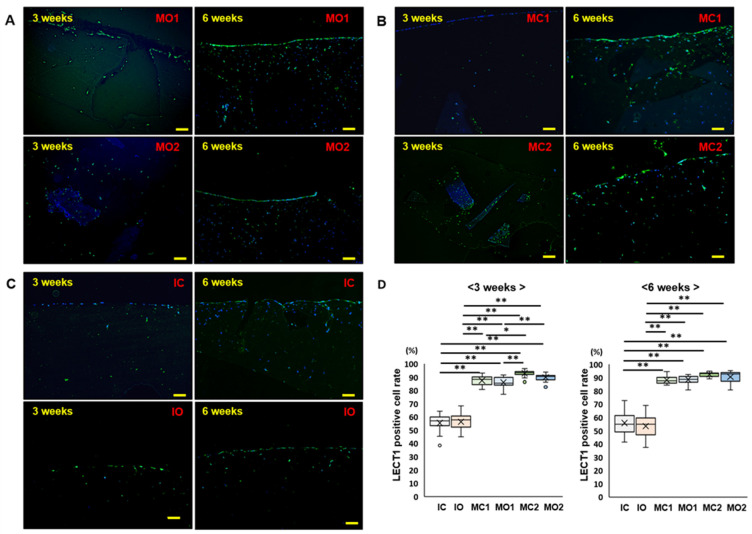
Immunohistochemistry of LECT1. (**A**): MO1 group at 3 and 6 weeks. (**B**): MO1 group at 3 and 6 weeks. (**C**): MO1 group at 3 and 6 weeks. Bar indicates 100 μm. (**D**): LECT1-positive cell rate in the atelocollagen gel at 3 and 6 weeks. The line of the box—median, ×—mean, small circle—outlier. N = 11 in each group. *—*p* < 0.05, **—*p* < 0.01.

**Figure 6 ijms-25-05563-f006:**
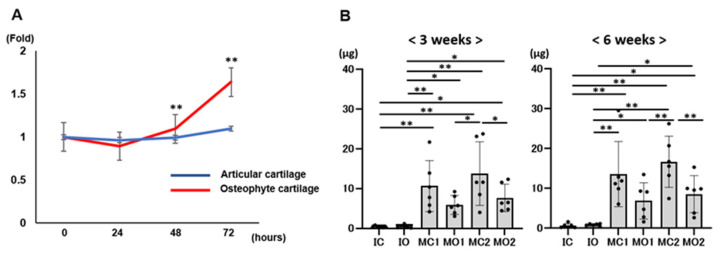
(**A**): Cell proliferation ability of isolated chondrocytes from osteophyte and articular chondrocytes. **—*p* < 0.01. (**B**): Analysis of glycosaminoglycan (GAG) contents in the composites at 3 and 6 weeks of culture. *—*p* < 0.05. **—*p* < 0.01.

**Table 1 ijms-25-05563-t001:** Results of RNA sequence comparing osteophyte and articular cartilages. Up- and down-regulated genes with padj < 0.05 and cartilage-specific genes. Bold: Statistically significant differences were observed at the assumed level of significance.

	GeneID	gene_sym	log2FoldChange (Osteophyte/Articular)	*p*-Value	padj
UP	**ENSG00000108821**	**COL1A1**	3.654	<0.01	0.034
	**ENSG00000134871**	**COL4A2**	3.32	<0.01	0.009
	**ENSG00000116774**	**OLFML3**	3.201	<0.01	0.009
	**ENSG00000176788**	**BASP1**	2.642	<0.01	0.012
down	**ENSG00000111341**	**MGP**	−1.676	<0.01	0.004
	**ENSG00000078596**	**ITM2A**	−1.805	<0.01	0.001
	**ENSG00000112562**	**SMOC2**	−2.004	<0.01	0.002
	**ENSG00000160307**	**S100B**	−2.273	<0.01	0.002
	**ENSG00000125845**	**BMP2**	−2.336	<0.01	0.002
no difference	ENSG00000139219	COL2A1	−0.083	0.891	0.9998
	ENSG00000157766	ACAN	−0.671	0.123	0.9998
	ENSG00000234899	SOX9-AS1	−0.239	0.855	0.9998

**Table 2 ijms-25-05563-t002:** Description of groups. All specimens were harvested from 11 patients. The osteophytes and articular cartilage of each patient were divided into six groups. The patients consisted of four men and seven women, with a mean age of 78.5 ± 4.6 (72–85) years.

Groups	Contents	Cell Number or Weight
IC	isolated chondrocyte from articular cartilage	2.0 × 10^5^
IO	isolated chondrocyte from osteophyte cartilage	2.0 × 10^5^
MC1	minced cartilage from articular cartilage	12.5 mg
MO1	minced cartilage from osteophyte cartilage	12.5 mg
MC2	minced cartilage from articular cartilage	25 mg
MO2	minced cartilage from osteophyte cartilage	25 mg

## Data Availability

The data presented in this study are available on request from the corresponding author. The data are not publicly available due to privacy and ethical reasons.

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
