# Peer review of "Osteophyte Cartilage as a Potential Source for Minced Cartilage Implantation: A Novel Approach for Articular Cartilage Repair in Osteoarthritis"

_ijms, 2024, doi:10.3390/ijms25105563_

Round 1

Reviewer 1 Report

Comments and Suggestions for Authors

Osteophyte cartilage as a potential source for minced cartilage implantation: A novel approach for articular cartilage repair in osteoarthritis

Suggestion: Minor revisions

1.      Line 65, please provide recent reference in addition to your current reference to support your statement.

2.      Line 86, please define IOD in full when it is mentioned for the first time.

3.      For your results section, please re-title all the sub-sections with the main result of that section in 7-10 words and not what you are comparing or observing. For example in section 2.1 -What was the major difference noticed that you discovered? 2.2 – what did the cultures reveal that is not already known?

4.      Are you able to provide any figures from your bulk RNA seq data with respect to any further analysis performed? Clusters? Volcano plots? Anything to add to your current manuscript? Would be good to see some more of the sequencing data too.

5.      For table 2 – please provide the number of patients in each group, followed the number of males and females in each of these groups. Also add the age range of the patients in each of these groups. I know these are outlined in the text, however adding these as a table will make it easier for your readers.

6.      In Figures 2 and 3 – please could you indicate in your figure legends what is indicated in the line of the box plots? Mean? Median? This is currently unclear.

7.      There is no figure 4 in the manuscript

8.      You may combine the figures 5 and 6 under one sub-title

9.      Please remove ‘*;p<0.5’ etc values from inside the actual figure panel. It is already mentioned in the figure legends.

10.   Please re-do figures 3 and 5 as figure 6 in box plots indicating how many patients (or ‘n’ ) were in each group.

11.   Did you perform any characterisation studies for the chondrocytes isolated to confirm that the cells you are using are indeed chondrocytes? This should have ideally been performed before performing any of your experiments. Could have been performed using flow cytometry or phenotyping, cell morphology – anything to confirm the cell type? If not – how do you justify that?

Comments on the Quality of English Language

Minor editing of English language required

Author Response

  1. Line 65, please provide recent reference in addition to your current reference to support your statement.

According to the reviewer’s comment, we added the following reference, “Jørgensen AEM, Agergaard J, Schjerling P, Heinemeier KM, van Hall G, Kjaer M. The regional turnover of cartilage collagen matrix in late-stage human knee osteoarthritis. Osteoarthr Cartil 2022;30(6):886-895.”.

  1. Line 86, please define IOD in full when it is mentioned for the first time.

We spelled out of IOD, line 88, “integrated optical density”.

  1. For your results section, please re-title all the sub-sections with the main result of that section in 7-10 words and not what you are comparing or observing. For example in section 2.1 -What was the major difference noticed that you discovered? 2.2 – what did the cultures reveal that is not already known?

According to the reviewer’s comment, we re-titled all the sub-sections with the main result as following, “2.1. Osteophyte cartilage has similar properties to articular cartilage”, “2.2. Matrix staining of osteophyte and articular cartilage fragments in atelocollagen gel is maintained after 6 weeks of culture”, “2.3. Osteophyte cartilage has better cell migration and proliferation abilities in the gel than articular cartilage”,” 2.4. Migrated and proliferated cells in the gel are chondrocytes from osteophyte cartilage”, and “2.5. Osteophyte chondrocytes have GAG production ability and better proliferation potential than articular chondrocytes”.

  1. Are you able to provide any figures from your bulk RNA seq data with respect to any further analysis performed? Clusters? Volcano plots? Anything to add to your current manuscript? Would be good to see some more of the sequencing data too.

We added the additional figures as new Figure 2

  1. For table 2 – please provide the number of patients in each group, followed the number of males and females in each of these groups. Also add the age range of the patients in each of these groups. I know these are outlined in the text, however adding these as a table will make it easier for your readers.

In this series, samples were harvested from 11 patients. Osteophyte and articular cartilage from each patient were divided into six groups. Therefore, all 6 groups are the same in number, age and sex. We added the following sentences in the figure legend, line 146-148, “All specimens were harvested from 11 patients. The osteophytes and articular cartilage of each patient were divided into 6 groups. The patients consisted of 4 men and 7 women, with a mean age of 78.5±4.6 (72-85) years”.

  1. In Figures 2 and 3 – please could you indicate in your figure legends what is indicated in the line of the box plots? Mean? Median? This is currently unclear.

We added the following sentences, line 157, 174, and 198, “The line of the box; median, ×; mean. N=11 in each group.”.

  1. There is no figure 4 in the manuscript

I added the new figure as Figure 5.

  1. You may combine the figures 5 and 6 under one sub-title.

We combined with 2 figures.

  1. Please remove ‘*;p<0.5’ etc values from inside the actual figure panel. It is already mentioned in the figure legends.

We removed *;p<0.05 from inside the figure.

  1. Please re-do figures 3 and 5 as figure 6 in box plots indicating how many patients (or ‘n’ ) were in each group.

As mentioned above, all groups consisted with 11 patients. Therefore, we added “N=11 in each group.” In figure legends.

  1. Did you perform any characterisation studies for the chondrocytes isolated to confirm that the cells you are using are indeed chondrocytes? This should have ideally been performed before performing any of your experiments. Could have been performed using flow cytometry or phenotyping, cell morphology – anything to confirm the cell type? If not – how do you justify that?

We did not perform any characterization studies for the chondrocytes isolated to confirm that the isolated cells were chondrocytes. Instead, we performed immunohistochemistry of LECT1 (chondromodulin) and confirmed the cells in the gel were stained with LECT1, which indicated that isolated cells were chondrocytes. We added Figure 5 which showed immunohistochemistry of LECT1.  

Reviewer 2 Report

Comments and Suggestions for Authors

General characteristics of the work:

The article Osteophyte Cartilage as a Potential Source for Minced Cartilage Implantation: A Novel Approach for Articular Cartilage Repair in Osteoarthritis investigates the feasibility of using osteophyte-derived cartilage for minced cartilage implantation (MCI) to treat osteoarthritis. Conducted through in vitro studies with samples from 22 patients, the research compares the regenerative capabilities of osteophyte and articular cartilage. Findings indicate that osteophyte cartilage shows higher chondrocyte proliferation and similar gene expression levels compared to articular cartilage, suggesting it as a viable source for MCI. While promising, the study highlights the need for further in vivo research to fully assess the clinical implications of this approach.

Overall, despite the relatively small population group, the work is interesting written correctly , but I recommend some minor corrections before moving on. My comments can be found below.

Minor commnets:

The beginning of the introduction requires a minor reconstruction. Please emphasize that cartilage damage along with other factors such as work, sports participation, musculoskeletal injuries, obesity and gender can influence the formation and progression of degenerative changes in the joint. Information about this, along with the necessary literature, should be added in the in the 1st paragraph of the introduction. Authors may find some useful information in the works: doi: 10.35784/acs-2023-40; DOI: 10.1056/NEJMcp1903768.

Figure 1, especially parts B and D are not very legible in my version of the paper please enlarge and adjust the fonts as recommended by the journal. It would be best to present each part separately.

In Table 1, it would be useful to use bold where statistically significant differences are observed at the assumed level of significance.

Figure 2 also needs reconstruction. Please change the color of the fonts, yellow on a light background is not very legible. Please also improve the quality of sections D and E.

Figure 3 as above for part B.

Figure 4 is missing or misnumbered. Please check the entire manuscript thoroughly.

The chapter on statistical analysis should be expanded and supplemented with literature.

After making the appropriate corrections and completing the content and literature, the article can be further processed and accepted for publication. I congratulate the authors on their interesting work.

Comments on the Quality of English Language

Minor editing of English language required.

Author Response

The article Osteophyte Cartilage as a Potential Source for Minced Cartilage Implantation: A Novel Approach for Articular Cartilage Repair in Osteoarthritis investigates the feasibility of using osteophyte-derived cartilage for minced cartilage implantation (MCI) to treat osteoarthritis. Conducted through in vitro studies with samples from 22 patients, the research compares the regenerative capabilities of osteophyte and articular cartilage. Findings indicate that osteophyte cartilage shows higher chondrocyte proliferation and similar gene expression levels compared to articular cartilage, suggesting it as a viable source for MCI. While promising, the study highlights the need for further in vivo research to fully assess the clinical implications of this approach.

Overall, despite the relatively small population group, the work is interesting written correctly , but I recommend some minor corrections before moving on. My comments can be found below.

Minor commnets:

The beginning of the introduction requires a minor reconstruction. Please emphasize that cartilage damage along with other factors such as work, sports participation, musculoskeletal injuries, obesity and gender can influence the formation and progression of degenerative changes in the joint. Information about this, along with the necessary literature, should be added in the in the 1st paragraph of the introduction. Authors may find some useful information in the works: doi: 10.35784/acs-2023-40; DOI: 10.1056/NEJMcp1903768.

According to the reviewer’s comment, we added the following sentence, line 33-34, “along with other factors such as work, sports participation, musculoskeletal injuries, obesity, and gender [1].”. and reference, “Sharma L. Osteoarthritis of the knee. N Eng J Med 2021;384(1):51-59”

Figure 1, especially parts B and D are not very legible in my version of the paper please enlarge and adjust the fonts as recommended by the journal. It would be best to present each part separately.

We modified Figure 1B and 1D.

In Table 1, it would be useful to use bold where statistically significant differences are observed at the assumed level of significance.

We modified table 1 according to the reviewer’s comment and added the following sentence, “Bold: Statistically significant differences were observed at the assumed level of significance.”.

Figure 2 also needs reconstruction. Please change the color of the fonts, yellow on a light background is not very legible. Please also improve the quality of sections D and E.

We modified Figure 2 according to the reviewer’s comments.  

Figure 3 as above for part B.

We modified Figure 3.

Figure 4 is missing or misnumbered. Please check the entire manuscript thoroughly.

We added the new figure as Figure 5.

The chapter on statistical analysis should be expanded and supplemented with literature.

We expanded the chapter on statistical analysis, line 436-438, “The data are presented as mean ± standard deviation (mean ± SD). Statistical significance was set at p <0.05. Statistical analysis was performed in the GraphPad Prism 9.0 (San Diego, CA, USA).”.

After making the appropriate corrections and completing the content and literature, the article can be further processed and accepted for publication. I congratulate the authors on their interesting work.